# Performance and feasibility of self-microsampling of capillary blood and saliva for serological testing of SARS-CoV-2

Ivonne Morales[ID][1,2]*, Josh Bueggeln[1], Anna Denzler[3], Vera Sonntag-Buck[4], Kathleen Börner[4,5,6,7], Petr Chlanda[6,8], Lisa Koeppel[1], Andreas Deckert[2], Till Bärnighausen[2], Michael Knop[3,9], Claudia M. Denkinger[ID][1,7]

**1** Department of Infectious Disease and Tropical Medicine, Heidelberg University Hospital, Heidelberg, Baden-Württemberg, Germany, **2** Heidelberg Institute of Global Health (HIGH), Heidelberg University Hospital, Heidelberg, Baden-Württemberg, Germany, **3** Center for Molecular Biology of Heidelberg University (ZMBH), Heidelberg, Baden-Württemberg, Germany, **4** Center for Infectious Diseases, Virology, Medical Faculty, Heidelberg University Hospital, Heidelberg, Baden-Württemberg, Germany, **5** AskBio GmbH, Heidelberg, Baden-Württemberg, Germany, **6** BioQuant, Heidelberg University, Heidelberg, Baden-Württemberg, Germany, **7** German Center for Infection Research (DZIF), Partner Site Heidelberg, Heidelberg, Baden-Württemberg, Germany, **8** Schaller Research Groups, Department of Infectious Diseases, Virology, Heidelberg University Hospital, Heidelberg, Baden-Württemberg, Germany, **9** German Cancer Research Center (DKFZ), Heidelberg, Baden-Württemberg, Germany

* ivonne.morales@uni-heidelberg.de

## Abstract

### Background

Serology is a valuable tool to estimate infections, case-fatality rates, and immunity. However, venipuncture and clinical infrastructure hinder scalability. This study evaluated the performance, feasibility and user experience of using a microsampling device for self-collected capillary blood and saliva to determine total SARS-CoV-2 S RBD antibodies.

### Methods

It included 149 participants with (n = 48) or without (n = 101) a known history of SARS-CoV-2 infection and confirmed antibodies. Venous and capillary blood were self- or professionally collected from all, and saliva was self-collected by 46/48 participants with antibodies. The detection of SARS-CoV-2 S RBD antibodies in all sample types was tested using a high-throughput platform and agreement was calculated. Intra- and inter-rater agreement between serum and capillary blood in participants with an unknown antibody status was also assessed. Participants evaluated the device's user-friendliness through questionnaires.

### Results

Among the 48 participants with known past infection and antibodies, agreement was 100% (95% CI: 92.6–100) between serum and capillary blood (self-collected or

**Data availability statement:** All relevant data are within the paper and its Supporting Information files.

**Funding:** This study was financed by the Ministry of Science, Research and Arts, Baden-Württemberg and the German Federal Ministry for Education and Science (BMBF) via the University Medicine Network (NUM), Project B-FAST, grant number 01KX2021. P.C. acknowledges funding from the Chica and Heinz Schaller Foundation. The funders had no role in study design, data collection or analysis, decision to publish, or preparation of the manuscript.

**Competing interests:** The authors have declared that no competing interests exist.

professionally collected). Self-collected saliva had slightly lower agreement with paired serum samples (95.7%, CI: 85.2–99.5). For the 101 participants without prior evidence of antibodies or infection, serum and self-collected capillary blood had good intra-rater agreement and serum and professionally collected capillary blood had almost perfect intra-rater agreement. Inter-rater agreement was also almost perfect. While 81.8% found the self-finger prick easy, 53.4% found using the microsampler easy. Among those who collected saliva, 84.8% found capillary blood easier to collect compared to saliva (52.2%).

## Conclusions

Our results show that detecting SARS-CoV-2 antibodies from capillary blood and saliva collected with the VAMS microsampling device is feasible and yields valid results. To ensure accuracy and reliability, additional training in self-sampling techniques may be essential. The positive user experience further underscores the microsampling device's potential for scalable serosurveillance and strengthening pandemic preparedness efforts.

## Introduction

The severe acute respiratory syndrome coronavirus 2 (SARS-CoV-2) pandemic has highlighted the crucial role of diagnostics in guiding disease control efforts [1]. Serological assays provide an effective surveillance tool to more accurately estimate the burden of infection, the case-fatality rate, and immunity at the population level [2]. However, current antibody testing strategies for SARS-CoV-2 have limited scalability due to the requirement for venipuncture samples collected by healthcare professionals in a clinical setting, which is costly, time-consuming, and pose risks of virus exposure [3]. Therefore, less invasive and scalable sampling methods that allow for remote self-collection are essential for large scale serosurveillance and to reduce costs [4]. In this respect, blood microsampling technologies, such as the dried blood spot (DBS) and volumetric absorptive blood microsampling (VAMS) provide an advantage over traditional venipuncture by enabling less painful capillary blood collection through a finger prick [4]. While DBS is a well-established method for the screening of infant disorders and the diagnosis and serosurveillance of infectious diseases (e.g., HIV, HBV, HCV, and more recently, SARS-CoV-2) [5,6], VAMS is a more novel approach that is also efficacious for the measurement of antibodies against viruses, including influenza and SARS-CoV-2 [7–10]. Also, in contrast to the DBS, VAMS allows for the collection of a standard and precise volume of finger prick blood onto a polymeric tip using capillary forces, it is compatible with high-throughput platforms, is less cumbersome to process than DBS cards and minimizes hematocrit-related variations [7–12]. The Mitra device from Neoteryx LLC, which employs VAMS technology, has been used to detect antibodies to SARS-CoV-2 from capillary blood, showing comparable results to serum samples when used with the Roche Elecsys assay [7] or in an ELISA [13] or nanoimmunoassay format [9].

Saliva presents another promising, non-invasive alternative for antibody detection, showing ease of self-collection and sustained IgG levels of up to 15 months post SARS-CoV-2 infection [14,15]. Furthermore, saliva may serve as a potential serosurveillance specimen for SARS-CoV-2 or other pathogens [16] that could be scaled up if used in high-throughput platforms.

In this study, we aimed to evaluate the performance and feasibility of the Mitra VAMS device for self-collection of finger prick blood and saliva to measure anti-SARS-CoV-2 antibodies using a high-throughput electrochemiluminescence-based SARS-CoV-2 spike (S) antibody assay.

## Methods

### Study population and recruitment

Eligible participants were aged 18 or older, able to provide a venous blood sample, attend a clinic visit, and provide written informed consent. Participants were enrolled in the study between December 1, 2021 and February 17, 2022 by phone or e-mail from two previous studies conducted at Heidelberg University Hospital: a cross-sectional serosurvey (July-November 2020) on anti-SARS-CoV-2 antibodies in Heidelberg and the Rhein-Neckar District, where antibody status was unknown at enrollment into the current study, and a biomaterial donor study. Donors with a past positive SARS-CoV-2 PCR and lab-confirmed anti-SARS-CoV-2 IgG antibodies were invited to participate (S1 Appendix). The groups are referred to as the "unknown antibody status" (from the seroprevalence study) and "known antibody status" (from the biomaterial donor study). Written informed consent was obtained from all participants enrolled in the study.

### Capillary blood and venous blood collection

All 149 study participants received a CE-IVD certified capillary blood collection kit (Neoteryx LLC) at the study clinic. Each kit contained a Mitra cartridge with two 30 μl VAMS tips [17], a gauze, plasters, a pair of disposable lancets (2 mm x 1.5 mm), specimen bag with desiccant, an instructional leaflet in English and German (S2 Appendix—the leaflet shown was modified from the original and is provided here only for illustrative purposes) and a return envelope. At the study clinic, participants had a designated workspace equipped with the kit, instructions, and a laptop to play an instructional video. While self-sampling and completing a questionnaire, a health professional evaluated the quality of their samples and then proceeded to collect additional capillary blood with a microsampler, as well as 5 mL of venous blood in EDTA vacutainer tubes (Sarstedt AG & Co. KG, Germany).

### Saliva collection

Saliva was only requested from the 48 participants with known antibody status. They received a modified version of the manufacturer provided instructional leaflet (in English and German, S2 Appendix—the leaflet shown was modified from the original and is provided here only for illustrative purposes) and a plastic container marked to hold about 1 mL of saliva. They then sequentially filled two microsampler tips with the specimen.

### Sample storage

Capillary blood and saliva samples were dried in their cartridges within the specimen bag at room temperature (RT) (21–25°C) overnight and then stored at −80°C until further analysis. Venous blood samples were refrigerated at 4°-8°C overnight, then processed for serum and stored at −80°C.

### Sample processing

Venous blood samples were processed for serum by centrifugation at 3000 rpm for 10 minutes (min) the day after collection. On the day of sample extraction, cartridges were removed from the −80°C freezer and thawed at RT. Each

microsampler comes labeled with a barcode for unique sample identification. The volumetric tip was detached and placed in the bottom of a 2 mL 96-well plate. Each microsampling tip holds 30 µl, with approximately 15 µl estimated to correspond to dried serum [7]. Each well was subsequently filled with 420 µL of extraction buffer (1% BSA, 50mM Tris (pH 8.0), 140 mM NaCl, 0.05% Tween-20) [7] for a total volume of approximately 450 µl per well. Plates were shaken for 2 h at 600 rpm RT covered with adhesive foil (Thermo Fisher Scientific Cat# AB0626) to prevent evaporation. The same procedure was conducted for saliva and serum samples, except that 15 µl of intact serum (based on the blood to serum ratio considered above) were diluted into 435 µl of extraction buffer for a final volume of 450 µl.

### Antibody assay

We used the electrochemiluminescence-based Roche Elecsys anti-SARS-CoV-2 immunoassay. This is a quantitative assay that reports antibodies (including IgG) to the SARS-CoV-2 S RBD protein as U/ml. The cut-off point of 0.80 U/mL differentiates samples as positive (≥ 0.80 U/mL) or negative (< 0.80 U/mL). The values between 0.40–250 U/mL represent the measuring range. A 1:10 or 1:100 dilution in extraction buffer was conducted for samples above the measuring range (>250 U/ml). All sample results have been corrected for dilution factors to represent the concentration within the sample matrix [7]. The assay was run on a Roche Cobas 6000 e601 Immunoassay Analyzer according to the manufacturer's instructions. A minimum volume of approximately 330 µl of sample is required when using a tube insert. The manufacturer reported clinical sensitivity and specificity of this immunoassay are 98.8% (95% Confidence Interval [CI]: 98.1–99.3%) and 99.96% (95% CI: 99.91–100.0%), respectively [18].

### Sample collection video

Participants could choose to watch an online instructional video on self-administering a finger prick for capillary blood collection, use only the enclosed leaflet, or use both resources. A professionally filmed instructional video in German and English was developed for this study, based on existing videos from the Neoteryx website [19]. The English version of the professionally filmed video can be accessed through the following link: https://vimeo.com/651712968/4c38c3915e?share=copy.

### Observation form and participant questionnaires

The user experience (ease of use) with the device and acceptability of the self-sampling procedure were assessed through participant and observer questionnaires. A health professional observed the sample collection process and only intervened if the participant struggled to initiate self-sampling (e.g., unable to open the cartridge), which the observer documented. Participants completed questionnaires covering demographics, occupational status, educational background, prior experience with self-administered tests, SARS-CoV-2 vaccination status and infection history. They rated their user experience using a numerical Likert-type scale ranging from 1 ("very easy") to 5 ("very difficult"), while the observer assessed the participants' confidence to independently self-sample using a similar scale and recorded adherence to instructions as Yes/No. Confidence in a participant's ability to self-sample reflects the health professional's assessment of their competence based on their performance and dexterity.

### Statistics

All statistical analyses were conducted in R (version 4.2.1). Agreement with 95% CI (Clopper-Pearson) was calculated for serum, capillary blood, and saliva samples. It was defined as the positive percent agreement between paired serum and capillary blood or saliva samples. Since all participants with a known antibody status had positive serum samples, only the positive percent agreement was assessed. Intra-rater agreement evaluated capillary blood vs. serum in participants with unknown antibody status, with Cohen's kappa used to measure this agreement and interpreted as follows: < 0, poor;

0.00–0.20, slight; 0.21–0.40, fair; 0.41–0.6, moderate; 0.61–0.8, good; > 0.8, almost perfect [20]. Feasibility was defined as the proportion of participants with a self-collected sample producing a valid antibody test. Inter-rater agreement between participant and professional capillary blood samples was also assessed with Cohen's kappa, using the Kappa() function from the vcd package. Linear regression analyzed associations between log antibody titers and infection history, age, sex, and vaccine doses. Further analyses used linear regression to examine associations between age, sex, education, prior self-sampling/self-testing experience, or perceived ease of the sample collection method on sample collection time using the lm() function.

## Ethics statement

Written informed consent was obtained from all study participants. The study protocol was approved by the Ethical Review Board of the Medical Faculty of Heidelberg University Hospital (S-200/2021). All procedures were carried out in accordance with the ethical standards described in the Revised Declaration of Helsinki from 2013.

## Results

### Participant demographics

A total of 151 participants were enrolled in the study between November 2021 and February 2022 (Fig 1). One participant left before any study procedures and was excluded from the analysis. Another did not meet enrollment criteria and was also excluded. Of the remaining 149 participants, 101 (67.8%) had an unknown antibody status at enrollment, while 48

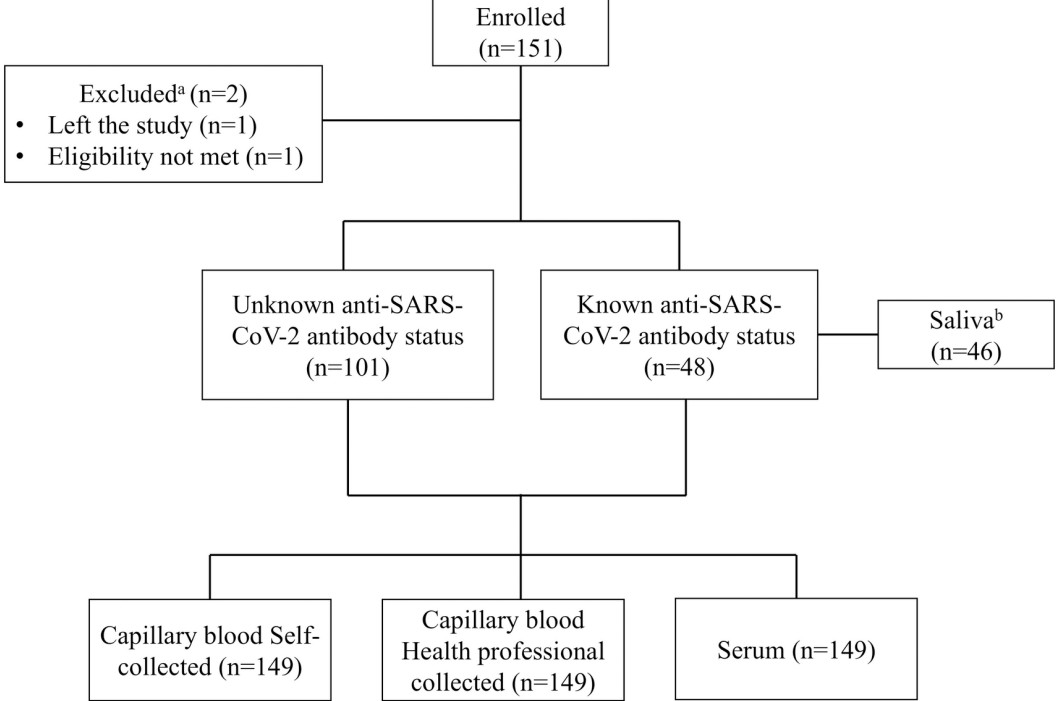

**Fig 1. Study flowchart.** a One study participant left the study before sample collection, and another did not fulfill the enrollment criteria and were excluded from all analyses. b Saliva was not collected from two study participants; correspondingly, no data on user experience, feasibility or prior use of saliva-based tests was obtained. However, both participants provided capillary and venous blood samples along with feedback on capillary blood collection. Hence, their demographic information, capillary blood-related questionnaires, and observer assessments were included in the analysis.

(32.2%) had a known antibody status. Blood samples were obtained from all 149 participants, and saliva samples were obtained from 46/48 with known antibody status.

Overall, there were more male (55.0%) than female participants (45.0%) and the median age was 53 years (interquartile range (IQR)= 25) (Table 1). A total of 98 out of 149 participants (65.8%) had completed upper secondary school while 49.0% (73/149) had a higher education degree. There was little previous experience with the use of an at-home SARS-CoV-2 test (6.7%) or other at-home tests (4.7%) requiring a self-administered finger prick. Nevertheless, 12.8% (19/149) of the participants had prior experience working in a laboratory, 20.0% (9/45) were familiar with SARS-CoV-2 saliva-based tests and 6.7% (3/45) regularly collected saliva at home (Table 1). While most participants were vaccinated against SARS-CoV-2 (95.3%), only 35.8% (53/148) reported a previous SARS-CoV-2 infection (Table 1). The two groups—those with a known antibody status and those with an unknown status—differed in age, lab working experience, number of vaccine doses and past SARS-CoV-2 infection (S1 Table).

### Agreement between paired serum and capillary blood or saliva samples

All professionally and self-collected capillary blood samples from participants with known antibody status (n = 48) tested positive for anti-S RBD antibodies, showing a positive percent agreement of 100% (CI: 92.6–100) with paired serum samples. In comparison, self-collected saliva samples had slightly lower positive percent agreement with paired serum samples at 95.7% (CI: 85.2–99.5) (Table 2).

Median antibody titers were highest in serum (32385 U/ml), followed by health professional-collected (25569 U/ml) and self-collected capillary blood (23788.5 U/ml). In saliva, the median titer was 60.5 U/ml (Fig 2) (S8 Table).

Multiple linear regression analysis showed a significant association between past SARS-CoV-2 infection and higher antibody titers in serum or capillary blood (S2–S3 Table). In the adjusted model, the number of vaccine doses was positively associated with higher antibody titers in self-collected capillary blood (S3 Table).

### Intra- and inter-rater agreement between paired capillary blood and serum samples

Among 101 participants with unknown antibody status, 97 (96.0%, CI: 89.6–98.7) tested positive for antibodies in both serum and self-collected capillary blood, and 96 (95.0%, CI: 88.3–98.2) tested positive in capillary blood collected by a health professional. Intra-rater agreement was better between serum and professionally collected capillary blood (κ = 0.88, CI: 0.66–1, $P < .001$) with 1 (1.0%) discordant sample, compared to self-collected capillary blood (κ = 0.74, CI: 0.39–1, $P < .001$) with 2 (2.0%) discordant samples (Table 3). Two of the three disagreements were from paired capillary samples (one self-collected, one professional) from the same participant, both testing negative (<0.4 U/ml). These samples were only tested in diluted form, and no original sample was available for retesting. The other discordant case was a low positive (5.25 U/ml) in self-collected capillary blood but negative in serum. Inter-rater agreement between self-collected and professionally collected capillary blood was almost perfect (κ = 0.88, CI: 0.66–1, $P < .001$) with one discordant sample, the same low positive noted above (Table 4).

### Feasibility and user experience

All participants successfully self-administered a finger prick, with 81.6% (120/147) using one lancet and 18.4% (27/147) using two (Fig 3). All 149 capillary blood samples, either self- or professionally collected produced valid antibody results. Common self-sampling deviations included not rubbing the hands together (37.4%, 55/147), not resting the hand on a firm surface (15.4%, 23/149), and not massaging the finger for blood flow (19.7%, 29/147). Additionally, 31.0% (45/145) did not wait two seconds before removing the microsampler tip from the blood drop (Fig 3). Other challenges included opening the cartridge (12.1%, 18/149) or the specimen bag (2.7%, 4/149), with one participant damaging the specimen bag. Additionally, 17 participants (11.4%) initially removed the sampler tips from the cartridge.

**Table 1. Participant demographic characteristics (N = 149).**

| | Participants |
|---|---|
| **Age, years** | |
| Median (IQR) | 53 (25) |
| **Sex, n (%)** | |
| Female | 67 (45.0%) |
| Male | 82 (55.0%) |
| **Highest general school qualification, n (%)** | |
| Upper secondary school | 98 (65.8%) |
| Intermediate secondary school | 38 (25.5%) |
| Lower secondary school | 12 (8.1%) |
| Other | 1 (0.7%) |
| **Vocational or technical training[a], n (%)** | |
| Yes | 91 (100.0%) |
| No | 0 (0.0%) |
| **Higher education degree, n (%)** | |
| Yes | 73 (49.0%) |
| No | 76 (51.0%) |
| **Previous experience with a SARS-CoV-2 test based on a self-administered finger prick, n (%)** | |
| Yes | 10 (6.7%) |
| No | 139 (93.3%) |
| **Regular use of an at-home medical test requiring a self-administered finger prick, n (%)** | |
| Yes | 7 (4.7%) |
| No | 142 (95.3%) |
| **Lab working experience, n (%)** | |
| Yes | 19 (12.8%) |
| No | 130 (87.2%) |
| **Previous experience with a saliva based SARS-CoV-2 test[b,c], n (%)** | |
| Yes | 9 (20.0%) |
| No | 36 (80.0%) |
| **Regular use of an at-home saliva collection device[b,c], n (%)** | |
| Yes | 3 (6.7%) |
| No | 42 (93.3%) |
| **Vaccinated against SARS-CoV-2[d], n (%)** | |
| Yes | 141 (95.3%) |
| No | 7 (4.7%) |
| **Number of vaccine doses, n (%)** | |
| 1 | 12 (8.5%) |
| 2 | 90 (63.8%) |
| 3 | 39 (27.7%) |
| **Past SARS-CoV-2 infection[e], n (%)** | |
| Yes | 53 (35.8%) |
| No | 95 (64.2%) |

*(Continued)*

**Table 1.** (Continued)

| | Participants |
|---|---|
| **Native language, n (%)** | |
| German | 139 (93.3%) |
| English | 0 (0.0%) |
| Other | 10 (6.7%) |

[a] Missing, n=58.

[b] Out of 46 participants that provided a saliva sample.

[c] Missing, n=1.

[d] Missing, n=1.

[e] Missing, n=1.

IQR = interquartile range.

**Table 2. Positive percent agreement between paired serum and capillary blood, or saliva samples from individuals with known antibody status.**

**Roche Elecsys anti-SARS-CoV-2 S RBD**

| | | Serum (n=48) | | | |
|---|---|---|---|---|---|
| **Known antibody status** | | **Positive** | **Negative** | **Total** | **Positive percent (%) agreement** |
| **Health professional capillary blood (n=48)[a]** | **Positive** | 48 | 0 | 48 | 100 (92.6-100) |
| | **Negative** | 0 | 0 | 0 | |
| | **Total** | 48 | 0 | 48 | |
| **Participants capillary blood (n=48)[b]** | **Positive** | 48 | 0 | 48 | 100 (92.6-100) |
| | **Negative** | 0 | 0 | 0 | |
| | **Total** | 48 | 0 | 48 | |
| **Participants saliva (n=46)[c]** | **Positive** | 44 | 0 | 44 | 95.7 (85.2-99.5) |
| | **Negative** | 2 | 0 | 2 | |
| | **Total** | 46 | 0 | 46 | |

[a] Samples collected by a health professional.

[b] Samples self-collected by participants.

[c] Saliva specimens were obtained from 46 of the 48 participants with known antibody status. The respective, 46 paired serum samples were used to measure positive percent agreement.

For saliva, all 46 participants provided a sample with a valid antibody result. Frequent deviations were not washing hands after collection (93.3%, 42/45), not abstaining from eating/drinking (39.1%, 18/46), and not filling tips sequentially (45.2%, 19/42) (Fig 4).

Confidence in self-sampling was high, especially for saliva (S1 Fig). Among capillary blood self-samplers, 77.2% received a confidence score of 4 (34.2%, 51/149) or 5 (43.0%, 64/149), while over 90% of saliva self-samplers received similar scores. Low confidence (6%, 9/149) was primarily due to issues with opening the cartridge or administering the self-fingerpick before preparing materials.

Overall, 146/149 (98.0%) of participants were perceived capable of independent capillary blood self-sampling. This was 'questionable' in three participants due to difficulties with opening the cartridge, the use of the lancet, the use of the micro-sampler tips, or filling of the tips. None had prior self-testing experience. All 46 (100%) participants who contributed saliva were considered capable of independent collection.

Participants found capillary blood collection generally easy, especially with video instructions, while saliva collection posed slightly more challenges (S1 Appendix and S2–S3 Figs). Suggested improvements included simplifying instructions

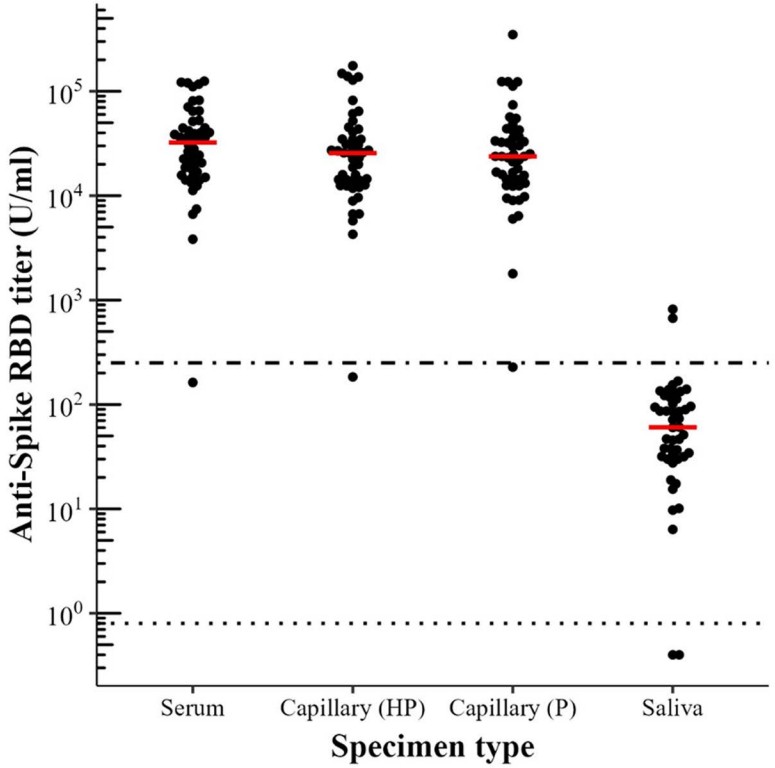

**Fig 2. Anti-S RBD antibody titers in serum, capillary blood, and saliva.** Antibody titer levels detected with the Elecsys anti-SARS-CoV-2 S RBD immunoassay in serum (n = 48), capillary blood (n = 48), and saliva (n = 46) from participants with known antibody status. The titers shown consider the sample dilutions from the extraction (1:30 for serum and capillary blood and 1:15 for saliva) as well as additional dilutions (1:10 to 1:1000) for samples above the measuring range (>250 U/ml). The red bar denotes the median antibody titer value. The dot-dashed line indicates the upper threshold for the measuring range (250 U/ml) and the dotted line denotes the manufacturer defined cut-off for positivity (0.8 U/ml). Capillary = capillary blood; HP = Health Professional; P = Participants.

**Table 3. Intra-rater agreement between paired capillary blood and serum samples from participants with unknown antibody status.**

| | | Serum | | | |
|---|---|---|---|---|---|
| **Capillary blood** | | **Positive** | **Negative** | **Total** | **Cohen's kappa** |
| **Health professional** | **Positive** | 96 | 0 | 96 | κ = 0.88 (CI: 0.66–1), *P* < .001 |
| | **Negative** | 1 | 4 | 5 | |
| | **Total** | 97 | 4 | 101 | |
| **Participants** | **Positive** | 96 | 1 | 97 | κ = 0.74 (CI: 0.39–1), *P* < .001 |
| | **Negative** | 1 | 3 | 4 | |
| | **Total** | 97 | 4 | 101 | |

and clarifying kit contents. Time required for sample collection was statistically significantly associated with the perceived ease of microsampler use and saliva collection by laypersons (S4–S7 Tables).

## Discussion

This study demonstrated comparable results for antibody detection using self-collected finger prick blood and saliva with a microsampling device. When combined with the Roche Elecsys platform, positive percent agreement between paired

**Table 4. Inter-rater agreement between paired capillary blood samples collected by participants or by a health professional.**

| Capillary blood | | Participants | | | Cohen's kappa |
|---|---|---|---|---|---|
| | | Positive | Negative | Total | |
| Health professional | Positive | 96 | 0 | 96 | κ = 0.88 (CI: 0.66–1), *P* < .001 |
| | Negative | 1 | 4 | 5 | |
| | Total | 97 | 4 | 101 | |

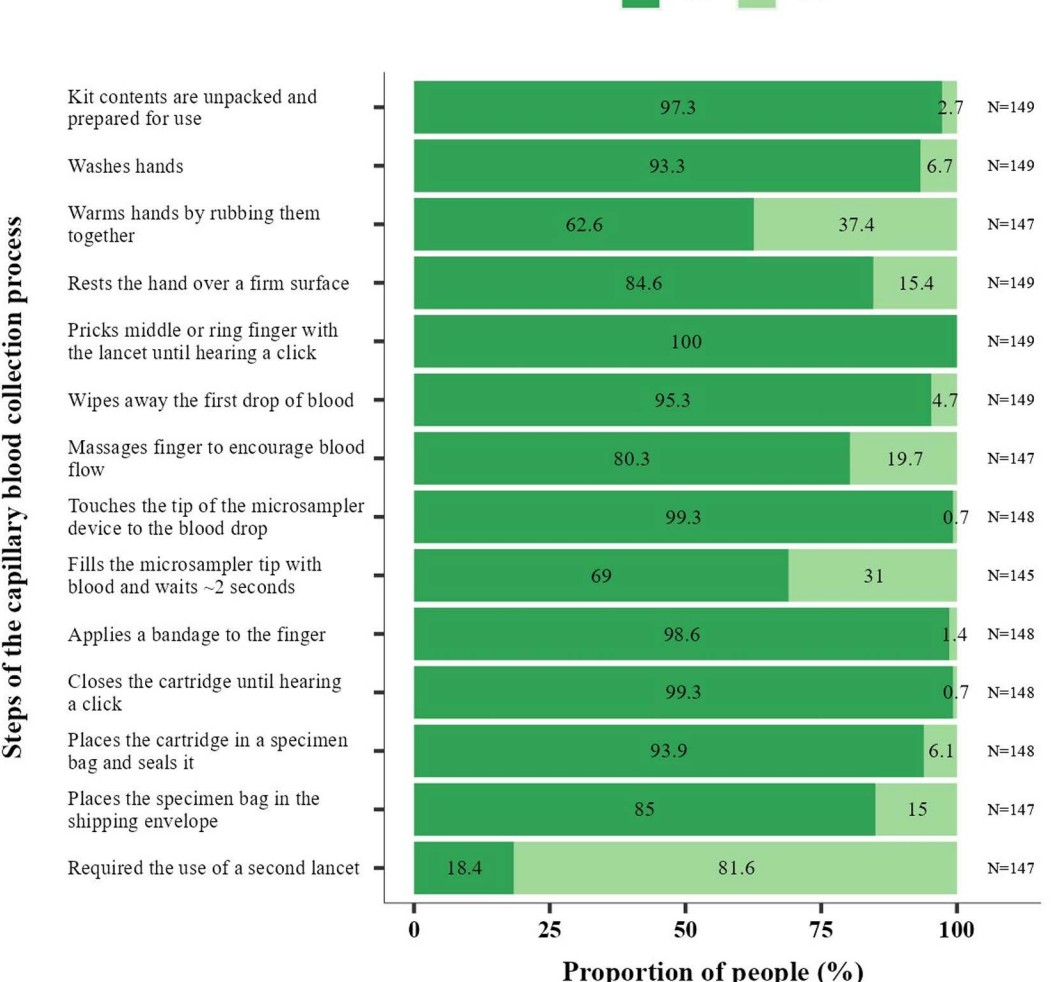

**Fig 3. Participant performance in capillary blood self-sampling.** Proportion of participants who completed each step of the self-sampling procedure. The observer assessed whether participants completed each step of the self-sampling procedure with a Yes or No response. N indicates the total number of participants observed.

serum and capillary blood samples collected by participants or a health professional was high (100%, CI: 92.6–100.0), consistent with previous findings by Garcia-Beltran et al. [7]. Although saliva showed slightly lower positive percent agreement with paired serum samples (95.7%, CI: 85.2–99.5), it supports the use of saliva with the Roche Elecsys SARS-CoV-2 S immunoassay [21].

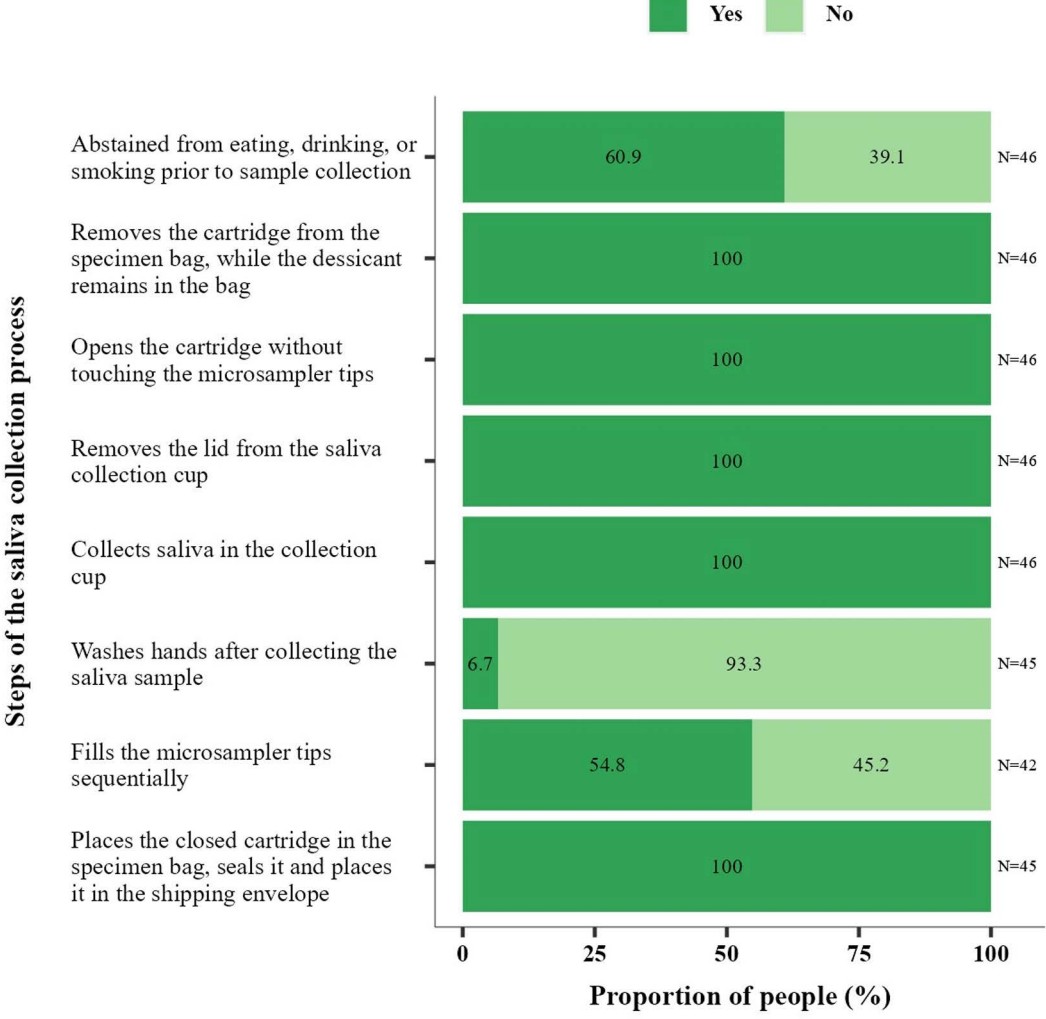

**Fig 4. Participant performance in saliva self-sampling.** Proportion of participants who completed each step of the self-sampling procedure. The observer assessed whether participants completed each step of the self-sampling procedure with a Yes or No response. N indicates the total number of participants observed.

While median antibody titers were slightly higher in serum compared to capillary blood, titers in saliva were approximately 1000-fold lower than in serum. This aligns with other studies reporting that salivary IgG concentrations can be considerably lower than in plasma or serum [14,15,22]. Additionally, deviations from saliva collection could have contributed to the overall lower antibody titers. Indeed, it has been suggested that salivary antibody concentrations may be affected by the salivary flow rate [23] or method compliance, such as abstaining from eating or drinking [14,24]. Thus, our findings highlight the importance of identifying potential pitfalls in self-sampling to emphasize compliance with collection procedures. Regression analysis suggested higher antibody levels in individuals with prior infection or vaccination.

We observed significant intra-rater agreement between capillary blood and serum samples in participants with unknown antibody status, ranging from good ($\kappa=0.74$) for those collected by study participants to almost perfect ($\kappa=0.88$) for professionally collected samples. These findings emphasize the importance of additional training in self-sampling techniques to ensure accuracy and reliability. Similarly, Garcia-Beltran et al. [7] reported high concordance (99%) between finger prick and serum samples.

Our study demonstrates high inter-rater agreement ($\kappa = 0.88$), confirming the viability of self-sampling by study participants. The one (1.0%) sample with discrepant results may be due to a false positive or potential sample contamination during processing.

We show that self-administered finger prick and saliva collection with the microsampling device is feasible [7,8,14]. Notably, all participants were able to provide finger prick blood or saliva samples yielding valid antibody results despite collection deviations. Positive participant experiences with the procedure and the device were reported. However, fewer participants (52.2%) found saliva collection easy compared to capillary blood collection (84.8%), potentially because the device was not originally designed for saliva. Nonetheless, using the same device for both sample types streamlines processing and adds convenience. Participant feedback on the saliva collection process suggested that a color indicator or marker could be useful to help recognize when the tips are full. Despite the challenges reported, more participants (51.1%) perceived saliva collection as easier for laypersons to learn compared to the self-finger prick (24.4%).

Limitations include a small sample size primarily composed of individuals with prior SARS-CoV-2 infection or vaccination, which may overestimate agreement and limit generalizability. The study population mostly consisted of individuals with an upper secondary degree, potentially affecting feasibility and user experience. Future studies should address variability in saliva collection and consider standardized protocols for more reliable results [25]. While antibody stability in VAMS-collected saliva remains less studied than in capillary blood, VAMS has proven effective for detecting stimulant drugs in oral fluids even under varying temperature, humidity, and light conditions [26]. As suggested in this study, VAMS application in the detection of antibodies from saliva could provide a future perspective of implementation in serosurveillance. We propose that antibody detection from saliva could be improved by extracting more than one microsampler tip, using a larger volume tip, and specimen dilutions could be optimized. Alternatively, other Roche Cobas platforms, such as the e801 analyzer which only requires a working volume of 200 µl, may be more suitable for analyzing specimens with low antibody concentrations [7].

While the VAMS-based microsampling device presents a promising alternative to venipuncture, a direct comparison with other conventional microsampling methods, such as DBS, could yield valuable insights, and future research assessing its cost-effectiveness would be beneficial. Building on strong evidence supporting the feasibility of at-home remote sampling using the microsampling device [8,14], our study uniquely enabled close monitoring of correct device usage and provided valuable feedback to enhance acceptance of the device and the self-sampling methodologies.

## Conclusions

In conclusion, we demonstrate that SARS-CoV-2 S RBD antibodies can be detected in self-collected capillary blood and saliva microsamples using a VAMS device. Our semi-automated workflow and high-throughput platform streamline sample processing for monitoring humoral responses in large populations. Additionally, our saliva findings suggest potential for integration into sensitive immunoassays and supporting broader pandemic preparedness strategies.

## Supporting information

**S1 Appendix. Study population characteristics and saliva collection challenges.**
(DOCX)

**S2 Appendix. Blood sample and saliva collection instructions.**
(PDF)

**S1 Table. Participant demographic characteristics by antibody status.**
(DOCX)

**S2 Table. Univariate and multivariable linear regression analysis on log antibody titers measured in serum, a past SARS-CoV-2 infection, age, sex, and number of vaccine doses.**
(DOCX)

**S3 Table. Univariate and multivariable linear regression analysis on log antibody titers measured in capillary blood, a past SARS-CoV-2 infection, age, sex, and number of vaccine doses.**
(DOCX)

**S1 Fig. Observer ratings of participant confidence during capillary blood or saliva collection.**
(DOCX)

**S2 Fig. Participant rating of the capillary blood self-sampling procedure.**
(DOCX)

**S4 Table. Demographic factors associated with time to collection of a capillary blood sample.**
(DOCX)

**S5 Table. Difficulties encountered during capillary blood collection, or which could make collection difficult for others.**
(DOCX)

**S6 Table. Participants' feedback on capillary blood collection.**
(DOCX)

**S3 Fig. Participant rating of the saliva self-sampling procedure.**
(DOCX)

**S7 Table. Demographic factors associated with time to collection of a saliva sample.**
(DOCX)

**S8 Table. Antibody test results.**
(XLSX)

## Acknowledgments

We thank the participants that took part in the study and the team of the BioBank Med V study, as well as the Department of Infectious Disease and Tropical Medicine at the Heidelberg University Hospital for facilitating the study. For the publication fee we acknowledge financial support by Heidelberg University.

## Author contributions

**Conceptualization:** Ivonne Morales, Claudia M. Denkinger.

**Data curation:** Ivonne Morales, Josh Bueggeln, Anna Denzler.

**Formal analysis:** Ivonne Morales, Josh Bueggeln, Lisa Koeppel.

**Funding acquisition:** Andreas Deckert, Till Bärnighausen, Claudia M. Denkinger.

**Investigation:** Ivonne Morales, Josh Bueggeln, Anna Denzler.

**Methodology:** Ivonne Morales, Anna Denzler, Vera Sonntag-Buck, Kathleen Börner, Petr Chlanda, Lisa Koeppel, Michael Knop.

**Project administration:** Ivonne Morales, Claudia M. Denkinger.

**Resources:** Vera Sonntag-Buck, Kathleen Börner, Petr Chlanda, Andreas Deckert, Till Bärnighausen, Michael Knop, Claudia M. Denkinger.

**Supervision:** Ivonne Morales.

**Validation:** Ivonne Morales.

**Visualization:** Ivonne Morales, Anna Denzler.

**Writing – original draft:** Ivonne Morales.

**Writing – review & editing:** Ivonne Morales, Josh Bueggeln, Anna Denzler, Vera Sonntag-Buck, Kathleen Börner, Petr Chlanda, Lisa Koeppel, Andreas Deckert, Till Bärnighausen, Michael Knop, Claudia M. Denkinger.

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
