## [Decision Letter · Decision Letter 0]

Dear Dr. Morales,

While Reviewer 2 has recommended "acceptance", he has also provided comments that need to be addressed to ensure the manuscript meets the highest standards of clarity and rigor. Additionally, we kindly ask you to reconsider the title of your manuscript to ensure it accurately reflects its content and findings and to satisfy all reviewer's comment. Please note that if your manuscript includes a video, it must be made publicly available upon publication. Kindly include a link to the video in the revised submission.

We look forward to receiving your revised manuscript and thank you for your efforts in improving the quality of your submission. Should you have any questions or require clarification, please do not hesitate to contact us.

We look forward to receiving your revised manuscript.

Kind regards,

Gheyath K. Nasrallah

Academic Editor

PLOS ONE

Journal Requirements:

2. We note that “S2_Appendix.pdf”  in your submission contain copyrighted images. All PLOS content is published under the Creative Commons Attribution License (CC BY 4.0), which means that the manuscript, images, and Supporting Information files will be freely available online, and any third party is permitted to access, download, copy, distribute, and use these materials in any way, even commercially, with proper attribution. For more information, see our copyright guidelines: http://journals.plos.org/plosone/s/licenses-and-copyright.

A.    You may seek permission from the original copyright holder of “S2_Appendix.pdf “ to publish the content specifically under the CC BY 4.0 license.

B. If you are unable to obtain permission from the original copyright holder to publish these figures under the CC BY 4.0 license or if the copyright holder’s requirements are incompatible with the CC BY 4.0 license, please either i) remove the figure or ii) supply a replacement figure that complies with the CC BY 4.0 license. Please check copyright information on all replacement figures and update the figure caption with source information. If applicable, please specify in the figure caption text when a figure is similar but not identical to the original image and is therefore for illustrative purposes only.

4. In the online submission form, you indicated that “The data that support the findings of this study are not openly available due to reasons of sensitivity and are available from the corresponding author upon reasonable request. Data are located in controlled and secure access data storage at Heidelberg University Hospital. Video requests will be considered on a case-by-case basis due to reasons of sensitivity.”

**Additional Editor Comments:**

The video should be made publicly available through a link attached to their manuscript upon publication.

Reviewers' comments:

Reviewer's Responses to Questions

**Comments to the Author**

1. Is the manuscript technically sound, and do the data support the conclusions?

Reviewer #1: Yes

Reviewer #2: Yes

Reviewer #3: Yes

2. Has the statistical analysis been performed appropriately and rigorously?

Reviewer #1: Yes

Reviewer #2: Yes

Reviewer #3: Yes

3. Have the authors made all data underlying the findings in their manuscript fully available?

Reviewer #1: No

Reviewer #2: Yes

Reviewer #3: Yes

4. Is the manuscript presented in an intelligible fashion and written in standard English?

Reviewer #1: Yes

Reviewer #2: Yes

Reviewer #3: Yes

Reviewer #1: Evaluation of the Manuscript "Sensitive and Feasible SARS-CoV-2 Antibody Detection from Blood and Saliva Self-Microsamples" Submitted to PLOS ONE for Publication Consideration

Volumetric absorptive microsampling (VAMS) has emerged as a valuable tool for serosurveillance and serodiagnosis during the COVID-19 pandemic. VAMS allows for remote, at-home sample collection, enabling large-scale serosurveillance studies that revealed higher SARS-CoV-2 seroprevalence than initially expected. This technique offers several advantages, including ease of use, minimal invasiveness, room temperature storage, and fixed volume absorption, which improves accuracy and reduces hematocrit effects.

Given the abundance of evidence demonstrating the use of VAMS in serosurveillance studies, including SARS-CoV-2, this study provides limited new information to the literature. Additionally, the sample size is too small, and most study participants are educated, as noted by the authors, which compromises the feasibility evaluation. Despite its limitations, this study may add useful information on the use of professional videos for sample collection using VAMS. Below are my major comments for the authors to consider to enhance the importance of their study.

Major Comments:

1. Title Revision: The title should reflect what the authors have done and found. I suggest modifying it to: Utility and Feasibility of Self-Sampling of Capillary Blood Using a Mitra Volumetric Absorptive Microsampling (VAMS) Device for Serological Testing of SARS-CoV-2.

2. Objective Clarification (Lines 64-66): The objective should be modified to accurately reflect the study's aim. The sensitivity pertains to the assay, which in this case is the electrochemiluminescence-based Roche Elecsys anti-SARS-CoV-2 immunoassay. If the authors used VAMS for saliva collection, the objective statement and title should reflect this. The discussion should also support this with relevant results, as limited data exist on the use of VAMS for saliva collection, transportation, and storage in serosurveillance studies.

3. Terminology Adjustment (Line 198): Instead of "Sensitivity of capillary, saliva, and serum samples," it should be "Agreement between saliva or capillary blood collected using the VAMS device," to align with the clarification above.

4. Comparison Enhancement: The study would have more impact if it compared different VAMS devices or compared VAMS-based capillary blood findings with conventional dried blood spots (DBS).

5. Manuscript Revision: For the manuscript to be considered for publication, the authors must address all the aforementioned remarks and rewrite their manuscript accordingly.

Reviewer #2: COVID-19, caused by severe acute respiratory syndrome coronavirus 2 (SARS-CoV-2), has posed a number of challenges to the medical and scientific community in terms of medical management and, in particular, diagnosis. The aim is to diagnose patients so that they can be managed and to establish epidemiological surveillance. In order to reach a wider target group, it was important to validate new diagnostic methods and strategies.

This is the background to the study "Sensitive and feasible SARS-CoV-2 antibody detection from blood and saliva self-microsamples", the main objective of which was to evaluate the sensitivity and feasibility of the Mitra VAMS device for self-collection of finger prick blood and saliva to measure anti-SARS-CoV-2 antibodies using a high throughput electrochemiluminescence-based SARS-CoV-2 spike (S) antibody assay.

Strengths and weaknesses

The research was approved by the ethics committee and all the participants provided written informed consent. The article is well organized and largely follows the journal's manuscript guidelines. Immunoassays for the detection and monitoring of SARS-CoV-2 have been shown to be effective in saliva in previous studies. The originality of this study is that it allows us to evaluate the combination of self-sampling tools calibrated on different matrices, saliva and blood, with sensitive serological tests (Roche Elecsys anti-SARS-CoV-2 immunoassay) and high-throughput instruments (Roche Cobas 6000 e601).

We agree with the findings that this strategy would be effective in a pandemic context, where rapid diagnosis for early treatment is a key factor in epidemic control. This was an enormous challenge during the COVID-19 pandemic in a context of restricted circulation and individual safety.

However, “Agreement was better between serum and professionally collected capillary blood (κ=0.88, CI: 0.66-1, P<.001) with 1 (1.0%) discordant sample, compared to self-collected capillary blood (κ=0.74, CI: 0.39-1, P<.001) with 2 (2.0%) discordant samples”.

These results highlight the need for more training in the use of self-testing devices to ensure quality results. This needs to be emphasized in the text, although the data are available in the appendices.

Reviewer #3: I read the manuscript with great interest. The study focuses on the choice of sample and sampling methods, which are very critical in clinical laboratory diagnosis, in particular during a pandemic, quality laboratory diagnosis, and easy-to-collect samples are essential parts of pandemic containment. Thus, as part of preparedness for the coming pandemic, the study is important, despite the fact that they have employed an expensive technology that will make it difficult for resource-limited settings to benefit from the technology.

My concerns about the study:

1. The authors used high-throughput electrochemiluminescence technology, a very expensive one, for the antibody analysis. How is this feasible for scalability?

2. How could the authors talk about the cost-effectiveness of their method without comparing the prices of different versions of sample collection methods?

3. When it comes to result interpretation, my major concern comes from the study included small study participants, and they have a mixed population with natural infection and vaccination, which the authors also acknowledged. In addition to what they did, I expect the authors to disaggregate the study participants into three those with natural infection with no vaccination, those Vaccinated but not infected, and both infected and vaccinated, and re-do the analysis and see the test agreements between the groups.

**Do you want your identity to be public for this peer review?** For information about this choice, including consent withdrawal, please see our Privacy Policy

Reviewer #1: **Yes: ** Tesfaye Gelanew

Reviewer #2: No

Reviewer #3: No

---

## [Author Response · Author response to Decision Letter 1]

23 May 2025

Review Comments to the Author

Reviewer #1: Evaluation of the Manuscript "Sensitive and Feasible SARS-CoV-2 Antibody Detection from Blood and Saliva Self-Microsamples" Submitted to PLOS ONE for Publication Consideration

Volumetric absorptive microsampling (VAMS) has emerged as a valuable tool for serosurveillance and serodiagnosis during the COVID-19 pandemic. VAMS allows for remote, at-home sample collection, enabling large-scale serosurveillance studies that revealed higher SARS-CoV-2 seroprevalence than initially expected. This technique offers several advantages, including ease of use, minimal invasiveness, room temperature storage, and fixed volume absorption, which improves accuracy and reduces hematocrit effects.

Given the abundance of evidence demonstrating the use of VAMS in serosurveillance studies, including SARS-CoV-2, this study provides limited new information to the literature. Additionally, the sample size is too small, and most study participants are educated, as noted by the authors, which compromises the feasibility evaluation. Despite its limitations, this study may add useful information on the use of professional videos for sample collection using VAMS. Below are my major comments for the authors to consider to enhance the importance of their study.

Major Comments:

1. Title Revision: The title should reflect what the authors have done and found. I suggest modifying it to: Utility and Feasibility of Self-Sampling of Capillary Blood Using a Mitra Volumetric Absorptive Microsampling (VAMS) Device for Serological Testing of SARS-CoV-2.

We thank the reviewer for this helpful suggestion and agree that the title merits a revision to reflect a major aspect of this study. Since clinical utility has a specific meaning within the diagnostic value cycle—referring to a diagnostic test’s usefulness in improving patient outcomes, informing clinical decision-making, and optimizing healthcare resources—we prefer to use the term performance instead of utility.

The manuscript title has been changed to ‘Performance and feasibility of self-microsampling of capillary blood and saliva for serological testing of SARS-CoV-2.’

2. Objective Clarification (Lines 64-66): The objective should be modified to accurately reflect the study's aim. The sensitivity pertains to the assay, which in this case is the electrochemiluminescence-based Roche Elecsys anti-SARS-CoV-2 immunoassay. If the authors used VAMS for saliva collection, the objective statement and title should reflect this. The discussion should also support this with relevant results, as limited data exist on the use of VAMS for saliva collection, transportation, and storage in serosurveillance studies.

We thank the reviewer for this comment. We have included ‘saliva’ in the revised study title. Lines 77-79 include saliva as part of the objectives of the study: “In this study, we aimed to evaluate the performance and feasibility of the Mitra VAMS device for self-collection of finger prick blood and saliva to measure anti-SARS-CoV-2 antibodies using a high-throughput electrochemiluminescence-based SARS-CoV-2 spike (S) antibody assay.”

Indeed, limited data exist regarding the use of VAMS with saliva as a sample. We have now extended our discussion on this matter to highlight that valid antibody results were also obtained in saliva samples collected with the microsampler device. We also note that using the same type of device for both sample types (saliva and capillary blood) offers convenience as it can streamline sample processing. Moreover, we add evidence indicating that while antibody stability at various storage conditions and temperatures has been shown for influenza antibodies in VAMS-collected capillary blood, there is evidence of the feasibility of VAMS with oral fluids for the detection of stimulant drugs even under varying temperature, humidity, and light conditions. Finally, we extend this discussion with additional suggestions on how to improve antibody detection from saliva. This has been revised in the manuscript discussion (lines: 310-312; 315-320; 331-332; 335-339; 343-352).

3. Terminology Adjustment (Line 198): Instead of "Sensitivity of capillary, saliva, and serum samples," it should be "Agreement between saliva or capillary blood collected using the VAMS device," to align with the clarification above.

We thank the reviewer for the valuable comment and have adjusted the section title to ‘Agreement between paired serum and capillary blood or saliva samples.’ The term sensitivity has also been adjusted to ‘agreement’ or ‘positive percent agreement’ throughout the manuscript text and it is shown in a revised version of manuscript Table 2.

Table 2. Positive percent agreement between paired serum and capillary blood, or saliva samples from individuals with known antibody status.

Roche Elecsys anti-SARS-CoV-2 S RBD

Serum (n=48)

Known antibody status Positive Negative Total Positive percent (%) agreement

Health professional

capillary blood (n=48)a Positive 48 0 48 100 (92.6-100)

Negative 0 0 0

Total 48 0 48

Participants

capillary blood (n=48)b Positive 48 0 48 100 (92.6-100)

Negative 0 0 0

Total 48 0 48

Participants

saliva (n=46)c Positive 44 0 44 95.7 (85.2-99.5)

Negative 2 0 2

Total 46 0 46

a Samples collected by a health professional.

b Samples self-collected by participants.

c Saliva specimens were obtained from 46 of the 48 participants with known antibody status. The respective, 46 paired serum samples were used to measure positive percent agreement.

To help distinguish this agreement from the use of Cohen’s kappa, we have further distinguished between intra- and inter-rater agreement and this distinction has also been specified in the methods section.

In addition, since sensitivity/specificity are not assessed in this study, the following sentence was removed from the discussion ‘Pre-pandemic or negative patient samples were unavailable for this study; therefore, specificity could not be assessed.’

4. Comparison Enhancement: The study would have more impact if it compared different VAMS devices or compared VAMS-based capillary blood findings with conventional dried blood spots (DBS).

We thank the reviewer for this comment. While our study did not include DBS due to this study’s specific objectives and unavailable instrumentation, we recognize the importance of comparing microsampling techniques. Studies have highlighted key differences, particularly DBS's high variability in sample volume, which limits its utility for quantitative antibody measurements (1). Additionally, DBS workflows require manual or semi-automated processing steps, such as sample punching, centrifugation, and buffer extraction, increasing complexity and cost (2). In contrast, VAMS allows precise, standardized blood collection via capillary action (3), streamlining analysis by eliminating additional processing steps. Its compatibility with high-throughput platforms in the context of antibody detection has been shown (1, 4, 5), while minimizing hematocrit-related variations (3). We have added this background information to the manuscript introduction section, lines: 60-71. Additionally, we have included in the discussion a reflection for future research aimed at comparing VAMS with DBS, lines 348-350: ‘While the VAMS-based microsampling device presents a promising alternative to venipuncture, a direct comparison with other conventional microsampling methods, such as DBS, could yield valuable insights, and future research assessing its cost-effectiveness would be beneficial.’

We appreciate the reviewer’s insights and hope this clarification is helpful.

5. Manuscript Revision: For the manuscript to be considered for publication, the authors must address all the aforementioned remarks and rewrite their manuscript accordingly.

We thank the reviewer for the comments provided. We have tried to address the reviewer’s comments to the best of our ability and have adapted the manuscript accordingly.

Reviewer #2: COVID-19, caused by severe acute respiratory syndrome coronavirus 2 (SARS-CoV-2), has posed a number of challenges to the medical and scientific community in terms of medical management and, in particular, diagnosis. The aim is to diagnose patients so that they can be managed and to establish epidemiological surveillance. In order to reach a wider target group, it was important to validate new diagnostic methods and strategies.

This is the background to the study "Sensitive and feasible SARS-CoV-2 antibody detection from blood and saliva self-microsamples", the main objective of which was to evaluate the sensitivity and feasibility of the Mitra VAMS device for self-collection of finger prick blood and saliva to measure anti-SARS-CoV-2 antibodies using a high throughput electrochemiluminescence-based SARS-CoV-2 spike (S) antibody assay.

Strengths and weaknesses

The research was approved by the ethics committee and all the participants provided written informed consent. The article is well organized and largely follows the journal's manuscript guidelines. Immunoassays for the detection and monitoring of SARS-CoV-2 have been shown to be effective in saliva in previous studies. The originality of this study is that it allows us to evaluate the combination of self-sampling tools calibrated on different matrices, saliva and blood, with sensitive serological tests (Roche Elecsys anti-SARS-CoV-2 immunoassay) and high-throughput instruments (Roche Cobas 6000 e601).

We agree with the findings that this strategy would be effective in a pandemic context, where rapid diagnosis for early treatment is a key factor in epidemic control. This was an enormous challenge during the COVID-19 pandemic in a context of restricted circulation and individual safety.

However, “Agreement was better between serum and professionally collected capillary blood (κ=0.88, CI: 0.66-1, P<.001) with 1 (1.0%) discordant sample, compared to self-collected capillary blood (κ=0.74, CI: 0.39-1, P<.001) with 2 (2.0%) discordant samples”.

These results highlight the need for more training in the use of self-testing devices to ensure quality results. This needs to be emphasized in the text, although the data are available in the appendices.

We thank the reviewer for pointing this out. We have revised manuscript lines 322-326 to highlight the need for training in self-sampling techniques. The text now reads as follows: “We observed significant intra-rater agreement between capillary blood and serum samples in participants with unknown antibody status, ranging from good (�=0.74) for those collected by study participants to almost perfect (�=0.88) for professionally collected samples. These findings emphasize the importance of additional training in self-sampling techniques to ensure accuracy and reliability.” Additionally, we added to the abstract conclusions the following sentence: ‘To ensure accuracy and reliability, additional training in self-sampling techniques may be essential.’

Reviewer #3: I read the manuscript with great interest. The study focuses on the choice of sample and sampling methods, which are very critical in clinical laboratory diagnosis, in particular during a pandemic, quality laboratory diagnosis, and easy-to-collect samples are essential parts of pandemic containment. Thus, as part of preparedness for the coming pandemic, the study is important, despite the fact that they have employed an expensive technology that will make it difficult for resource-limited settings to benefit from the technology.

My concerns about the study:

1. The authors used high-throughput electrochemiluminescence technology, a very expensive one, for the antibody analysis. How is this feasible for scalability?

We thank the reviewer for this comment. The Roche Elecsys platform is a well-established platform with excellent performance, particularly suited for our exploratory work around alternative sample types. The Mitra microsampler has been demonstrated to be compatible with various high-throughput alternatives that may be less costly, such as in-house ELISA and nano-immunoassays. For example, Kalish et al. successfully used the Mitra microsampler with an in-house ELISA to determine SARS-CoV-2 positivity in nearly 10,000 individuals in the U.S (4, 6). Similarly, Michielin et al. utilized a nano-immunoassay capable of processing up to 1,024 samples in parallel with minimal sample and reagent consumption, showing promise for cost-effective serological testing (5). Additionally, the Mitra microsampler has been validated for use in influenza studies with an in-house bead-based multiplex assay, further supporting its scalability for large-scale applications using alternative high-throughput platforms (1).

2. How could the authors talk about the cost-effectiveness of their method without comparing the prices of different versions of sample collection methods?

We thank the reviewer for this comment. We would like to clarify that our study does not claim cost-effectiveness, as this was not within the scope of our research. Further studies are needed to properly evaluate cost-effectiveness. However, we do recognize the potential for cost-saving.

In large-scale serosurveillance programs, cost-effectiveness extends beyond the direct cost of sample collection to include factors such as sample preparation, shipping, storage, and analysis (3, 5, 7). The method we discuss offers significant advantages in these areas, principally over venipuncture, including eliminating the need for specialized personnel for sample collection, cold-chain storage and shipping, reducing reagent consumption, and allowing seamless integration with semi-automated bioanalytical laboratory workflows (3). In this regard, VAMS may also present an advantage over other microsampling methods, such as dried blood spots, which typically require additional specialized equipment—such as a puncher, extra processing workstations, and a compatible immunoanalyzer platform, as demonstrated by Beyerl et al. (2).

Furthermore, high-throughput capacity further enhances cost efficiency, particularly when paired with scalable serological assays. While the Roche Elecsys is relatively costly, VAMS has been explored with alternative platforms (see response to Q1). Moreover, as noted by Beyerl et al., (2) Roche Elecsys platforms are widely in use and benefit from a well-established supply chain coverage, making them accessible to centralized laboratories worldwide. Ultimately, cost-effectiveness should be evaluated in terms of the overall balance between cost and efficiency, rather than direct pricing alone.

We have added a sentence in the discussion (lines 353-355) to acknowledge the value of conducting a cost-effectiveness analysis in future research: ‘While the VAMS-based microsampling device presents a promising alternative to venipuncture, a direct comparison with other conventional microsampling methods, such as DBS, could yield valuable insights, and future research assessing its cost-effectiveness would be beneficial.’

3. When it comes to result interpretation, my major concern comes from the study included small study participants, and they have a mixed population with natural infection and vaccination, which the authors also acknowledged. In addition to what they did, I expect the authors to disaggregate the study participants into three those with natural infection with no vaccination, those Vaccinated but not infected, and both infected and vaccinated, and re-do the analysis and see the test agreements between the groups.

We thank the reviewer for this comment. The reviewer requests an analysis on 1) naturally infected without vaccination (n=4); 2) no natural infection with vaccination (n=92); and 3) naturally infected with vaccination (n= 48) (Table 1).

Table 1.

COVID-19 Vaccination status Total

Past COVID-19 infection No Yes Missing

No 3 92 0 95

Yes 4 48 1 53

Missing 0 1 0 1

Total 7 141 1 149

As reported in

---

## [Decision Letter · Decision Letter 1]

Performance and feasibility of self-microsampling of capillary blood and saliva for serological testing of SARS-CoV-2

PONE-D-24-58079R1

Dear Dr. Morales,

We’re pleased to inform you that your manuscript has been judged scientifically suitable for publication and will be formally accepted for publication once it meets all outstanding technical requirements.

Kind regards,

Ewurama Dedea Ampadu Owusu, PhD

Academic Editor

PLOS ONE

Additional Editor Comments (optional):

Reviewers' comments:

Reviewer's Responses to Questions

**Comments to the Author**

Reviewer #3: All comments have been addressed

2. Is the manuscript technically sound, and do the data support the conclusions?

Reviewer #3: Yes

3. Has the statistical analysis been performed appropriately and rigorously?

Reviewer #3: Yes

4. Have the authors made all data underlying the findings in their manuscript fully available?

Reviewer #3: Yes

5. Is the manuscript presented in an intelligible fashion and written in standard English?

Reviewer #3: Yes

Reviewer #3: (No Response)

**Do you want your identity to be public for this peer review?** For information about this choice, including consent withdrawal, please see our Privacy Policy

Reviewer #3: No

---

## [Editor Report · Acceptance letter]

PONE-D-24-58079R1

PLOS ONE

Dear Dr. Morales,

I'm pleased to inform you that your manuscript has been deemed suitable for publication in PLOS ONE. Congratulations! Your manuscript is now being handed over to our production team.

Kind regards,

on behalf of

Dr. Ewurama Dedea Ampadu Owusu

Academic Editor

PLOS ONE